# Major Surgical Trauma Impairs the Function of Natural Killer Cells but Does Not Affect Monocyte Cytokine Synthesis

**DOI:** 10.3390/life12010013

**Published:** 2021-12-22

**Authors:** Roman M. Müller-Heck, Björn Bösken, Ivo Michiels, Marcel Dudda, Marcus Jäger, Stefanie B. Flohé

**Affiliations:** 1Department of Trauma, Hand and Reconstructive Surgery, University Hospital Essen, 45147 Essen, Germany; roman.mueller@uk-essen.de (R.M.M.-H.); bjoern.boesken@rub.de (B.B.); ivo.michiels@uk-essen.de (I.M.); marcel.dudda@uk-essen.de (M.D.); 2Chair of Orthopedics and Trauma Surgery, University of Duisburg Essen, 45147 Essen, Germany; marcus.jaeger@uni-due.de; 3Department of Orthopedics, Trauma and Reconstructive Surgery, St. Marien Hospital Mülheim an der Ruhr, 45468 Mülheim, Germany

**Keywords:** surgery, trauma, injury, immune dysregulation, infection, natural killer cells, immunosuppression, monocytes, cytokines, marker

## Abstract

Major traumatic and surgical injury increase the risk for infectious complications due to immune dysregulation. Upon stimulation with interleukin (IL) 12 by monocyte/macrophages, natural killer (NK) cells release interferon (IFN) γ that supports the elimination of the pathogen. In the present study, we investigated the impact of invasive spine surgery on the relationship between monocytes and NK cells upon exposure to *Staphylococcus aureus*. Mononuclear cells and serum were isolated from peripheral blood of patients before and up to 8 d after surgery and stimulated with inactivated *S. aureus* bacteria. NK cell and monocyte function were determined by flow cytometry. NK cells continuously lost their ability to produce IFN-γ during the first week after surgery independently from monocyte-derived IL-12 secretion. IFN-γ synthesis was minimal on day 8 and was associated with decreased expression of the IL-12 receptor and activation of transcription factors required for *IFNG* gene transcription. Addition of recombinant IL-12 could at least partially restore NK cell function. Pre-operative levels of growth/differentiation factor (GDF) 15 in the serum correlated with the extent of NK cell suppression and with hospitalization. Thus, NK cell suppression after major surgery might represent a therapeutic target to improve the immune defense against opportunistic infections.

## 1. Introduction

Major traumatic or surgical tissue damage causes a dysregulation of the immune system and is associated with an increased risk for infectious complications [1]. Modulation of innate and adaptive immune cells has been reported but the underlying pathomechanisms are yet incompletely understood.

Human circulating NK cells consist of two major subpopulations: CD56^dim^ NK cells primarily act as cytotoxic effector cells that eliminate tumor cells and virus-infected cells and only weakly secrete cytokines [2]. In contrast, CD56^bright^ NK cells possess weak cytotoxic activity but release considerable amounts of diverse cytokines [3]. During bacterial infections, NK cells are a major source of IFN-γ that promotes the antimicrobial activity of macrophages and dendritic cells (DCs) [2,3,4]. Interleukin (IL) 12 is released by monocytes/macrophages and DCs upon contact with pathogen-derived components and activates NK cells for IFN-γ synthesis [5,6]. Thus, there exists a cross-talk between monocytes/macrophages/DCs and NK cells based on the positive feedback loop between IL-12 and IFN-γ.

The IL-12 receptor (IL12R) consists of a constitutively expressed β1 and an induced β2 chain. Binding of IL-12 to the IL-12R triggers the phosphorylation of signal transducer and activator of transcription (STAT) 4 that induces the transcription of the *IFNG* gene in the nucleus [7,8]. The T-box transcription factor T-bet cooperates with STAT4 in *IFNG* gene transcription and additionally promotes *IL12RB2* gene transcription in CD4^+^ T cells and likely in NK cells [9]. The cytokine IL-2 further drives the synthesis of IFN-γ by NK cells [10,11].

After severe traumatic injury, monocytes and NK cells are impaired in the release of IL-12 and IFN-γ, respectively, which disturbs the cross-talk between these cell types [12,13]. The suppression of NK cells is associated with a diminished IL-12R expression and signaling that is largely mediated by circulating factor(s) in the serum [13]. We identified one of these factors as growth/differentiation factor (GDF) 15, a member of the TGF-β family that is produced at extremely high levels after trauma especially in those patients who later develop a septic complication [13]. Moreover, at the end of the first week after trauma, NK cells are unable to produce IFN-γ independent from circulating factors and are refractory to the cytokines IL-2 and IL-12 that otherwise increase NK cell function [13]. Thus, the broad dysregulation of the immune system after severe trauma is associated with the disruption of the monocyte/NK cell cross-talk and includes cell-intrinsic changes and environmental signals.

The question arises whether the breakdown of the cellular cross-talk between monocytes and NK cells likewise occurs after surgical injury and if so whether there exists an early marker. To address this issue, we investigated the function of monocytes and NK cells from patients undergoing invasive spine surgery that is associated with major soft tissue damage and with an enhanced risk for infectious complications [14,15]. We demonstrate here that NK cells continuously lost their capacity to produce IFN-γ within the first week after surgery whereas monocyte function remained unaffected. NK cell suppression after surgery was associated with an altered expression of transcription factors involved in *IL12RB2* and *IFNG* gene transcription. We identified pre-operative levels of circulating GDF-15 as a potential marker for the extent of surgery-induced NK cell suppression and for delayed recovery of the patient.

## 2. Materials and Methods

### 2.1. Patients and Study Design

All patients underwent surgery at the Department of Orthopaedics and Trauma Surgery at the University Hospital Essen between April 2014 and September 2015 (for patient characteristics see Table 1). Inclusion criteria were Caucasian, age ≥ 15, and indication for long distance dorsal spine surgery due to scoliosis or degenerative central or lateral spinal stenosis. Minimal invasive procedures were not applicable. Exclusion criteria were immunosuppressive drug medication and pre-existing infections, or tumors.

In case of multilevel fusion for scoliosis, meticulous subperiosteal stripping and bone decortication (including the facet joints) were performed. Patients with spinal canal stenosis underwent laminectomy, partial facetectomy, and far lateral decompression of the segmental nerve roots. All patients additionally required instrumented fusion.

All interventions were performed by one and the same surgeon. Written informed consent of the patients and their parents in case of age < 18 was obtained before the first blood withdrawal. The study was conducted in accordance with the guidelines of the World Medical Association’s Declaration of Helsinki and was approved by the local ethic committee of the University Hospital Essen, Germany (13-5637-BO). Whole blood was drawn before surgery and 1, 4, and 8 d after surgery. Heparinized blood was used to isolate peripheral blood mononuclear cells (PBMC). Serum was obtained from coagulated blood.

### 2.2. Isolation and Culture of Mononuclear Cells

PBMC were isolated using the standard protocol for Ficoll density gradient centrifugation followed by erythrocyte lysis as described before [13]. Briefly, PBMC were cultured in “Very low endotoxin” VLE Roswell Park Memorial Institute (RPMI) 1640 Medium supplemented with 100 U/mL Penicillin, 100 µg/mL Streptomycin (Sigma-Aldrich, St. Louis, MO, USA), and 10% (*v*/*v*) serum. Cultures of PBMC were set up at least in triplicates (4 × 10^5^ cells/well) in 96-well flat bottom plates (BD Biosciences, Heidelberg, Germany) as described [13] and rested for 30 min. Where indicated, recombinant human IL-2 (1 ng/mL; Biotechne), IL-12 (1 ng/mL; Biotechne), or both was added before stimulation of the cells with heat-inactivated *S. aureus* (Pansorbin Cells Standardized, 0.05% (*v*/*v*), Calbiochem, Merck, Darmstadt, Germany) 30 min later. Unstimulated PBMC served as negative control. After 16 h, PBMC were harvested for further analyses.

In some experiments, PBMC from healthy donors were cultured in the presence of 4% (*v*/*v*) serum from patients. Therefore, the sera from all patients were pooled for each time point.

### 2.3. Flow Cytometry

Flow cytometry was performed as described previously [16]. Briefly, for the examination of the expression of surface molecules, PBMC (immediately after isolation or after culture) were stained with combinations of the following fluorochrome-conjugated antibodies: anti-human CD56 (clone CMSSB, eBio-Science, San Diego, CA, USA), anti-Human CD3 (clone MEM-57, Immuno-Tools, Friesoythe, Germany), anti-human IL12Rβ1 (clone 2.4E6, BD Biosciences, East Rutherford, NJ, USA), anti-human IL-12Rβ2 (clone 2B6/12β2, BD Biosciences), anti-human CD14 (clone MEM-15, Immuno-Tools). For intracellular detection of cytokines, PBMC were further cultured for 5 h in the presence of GolgiStop before fixation, permeabilization (Fixation and Permeabilization Solution, BD Biosciences), and staining of the cells for IFN-γ (clone 4S.B3, Biolegend) or IL-12p40 (clone 11.5, Biolegend, San Diego, CA, USA). For intracellular detection of transcription factors, PBMC were fixed and permeabilized (Foxp3 Staining Buffer Set, eBioScience™, Thermo Fischer Scientific, Waltham, MA, USA) and stained with anti-human T-bet (clone eBio4B10, eBioScience) or anti-human STAT4 (pY693) (clone 38/p-STAT4). For all antibodies, respective isotype control antibodies were used in separate samples to define the threshold of positive staining. Data were acquired with a FACScalibur (BD Biosciences) and analyzed with CellQuest Pro (BD Biosciences). NK cells were gated as CD3^−^CD56^bright^ in the SSC^lo^FSC^lo^ lymphocyte gate. Monocytes were gated as CD14^+^FSC^int^ cells.

### 2.4. Quantification of GDF-15

GDF-15 in the sera was quantified by ELISA (Thermo Fischer Scientific) as recommended by the manufacturer.

### 2.5. Statistical Analyses

For statistical evaluation of the data Prism 6.0 was used. Data are shown as individual values or as Tukey box plots as indicated. Significant differences between two data sets were tested using the Wilcoxon test. The significant association between two variables was examined using Spearman’s correlation. A two-tailed *p*-value < 0.05 was considered as significant.

## 3. Results

### 3.1. Major Spine Surgery Impairs the NK Cell Response to S. aureus

Twelve patients with elective major spine surgery met the inclusion criteria and were included in this pilot study. Surgery induced an increase in circulating leukocytes within 1 d. Moreover, elevated levels of the inflammatory markers IL-6 and C-reactive protein (CRP) were detected in the sera after surgery. In parallel, surgery caused a prominent increase of the concentration of myoglobin, an indicator of skeletal muscle damage, in the serum (Table 2).

The frequency of CD56^bright^ NK cells (gating of NK cells is shown in Figure 1a) in PBMC did not change after injury (Figure 1b). To mimic a potential contact of leukocytes with invading pathogens, PBMC were isolated before and at different time points after surgery and were exposed to inactivated *S. aureus* bacteria in vitro. Under these conditions, T cells did not produce IFN-γ at a relevant frequency [16]. FACS analysis of intracellular IFN-γ in CD56^bright^ NK cells from one representative patient before and 8 d after surgery is shown in Figure 1c. The frequency of IFN-γ-producing NK cells did not change within 1 d after surgery but strongly declined thereafter (Figure 1d).

The median suppression of IFN-γ synthesis on d4 after surgery was 70% and did not correlate with the age of the patients (Appendix A).

We have previously shown that the IFN-γ response from NK cells after stimulation of PBMC with *S. aureus* depends on IL-12 [16]. Monocytes release IL-12 in response to diverse microbial components and thus might play a role in the diminished activity of NK cells after surgery [5]. Therefore, the production of IL-12p40 (a subunit of bioactive IL-12) in monocytes before and after surgery was examined after stimulation of PBMC with *S. aureus*. The FACS analysis of intracellular IL-12p40 in monocytes from one representative patient before and 8 d after surgery is shown in Figure 2a. The frequency of IL-12p40-producing monocytes did not change after surgery at any time point (Figure 2b). The release of TNF-α, that is frequently used as an indicator of post-traumatic monocyte activity, was reduced on d8 after surgery but not at earlier time points (see Appendix A). Thus, NK cells lose their capacity to produce IFN-γ within 8 d after surgery despite unchanged IL-12 synthesis by monocytes.

### 3.2. Treatment with IL-12 Rescues NK Cell Activity after Surgery

We next investigated whether the NK cell activity could be restored after surgery. IL-12 and IL-2 that are known to boost NK cell-derived IFN-γ synthesis [5,10] and were added to PBMC during stimulation with *S. aureus*. IL-2 alone did not significantly increase the synthesis of IFN-γ by NK cells at any time point (Figure 3a and Appendix A). In contrast, IL-12 and the combination of IL-12 and IL-2 caused a significant increase in IFN-γ production before and after surgery (Figure 3b). Consequently, on day 4 the activity of NK cells was restored up to levels that were observed for stimulation with *S. aureus* before surgery. Notably, on day 1 after surgery the NK cells strongly responded to the supplemented cytokines and clearly exceeded the baseline activity of NK cells before surgery (Figure 3b).

To elucidate the question why the activity of NK cells declined beyond day 4 after surgery we determined the expression of the IL-12 receptor that is required to sense soluble IL-12. The expression of the constitutively expressed IL-12Rβ1 chain did not change after surgery (Appendix A). The expression of the inducible IL-12Rβ2 chain strongly declined on day 4 and even more on day 8 (Figure 4a,b) after injury but returned to pre-operative levels when IL-2 and IL-12 were added during the stimulation of PBMC with *S. aureus* (Figure 4b). As expected, the expression of the IL-12Rβ2 chain correlated with the production of IFN-γ (Figure 4c).

In order to get further insight into the origin of reduced IL-12Rβ2 expression after surgery we determined the expression of the transcription factors T-bet and STAT4 that regulate *IL12RB2* gene transcription. Stimulation of PBMC with *S. aureus* increased the frequency of T-bet^+^ NK cells (Figure 5a). On day 4 and on day 8 after surgery, NK cells displayed a significantly reduced capacity to enhance the expression of T-bet (Figure 5b). Addition of IL-2/IL-12 during stimulation with *S. aureus* increased the frequency of T-bet^+^ NK cells at day 8 after surgery (Figure 5c).

Moreover, stimulation of PBMC with *S. aureus* caused an increased activation of STAT4 in NK cells that was much weaker on day 8 after surgery (Figure 6a,b). In contrast to the expression of T-bet, the addition of IL-2/IL-12 did not change the frequency of pSTAT4^+^ NK cells (Figure 6c). Thus, addition of IL-12 rescues the activity of NK cells after surgery in conjunction with increased T-bet expression.

### 3.3. Circulating Factors Contribute to Late NK Cell Suppression

Increased concentration of circulating GDF-15 is involved in the suppression of NK cells after polytrauma [13] and might similarly play a role in the suppression of NK cells after major surgery. The levels of GDF-15 transiently increased by 2-fold within 1 d after surgery and declined thereafter (Figure 7a). There was no association between circulating GDF-15 after surgery with CRP or with myoglobin as a measure for surgery-induced skeletal muscle damage (Appendix A). Pre-operative GDF-15 levels inversely correlated with the capacity of NK cells to produce IFN-γ before and 4 d after surgery and positively correlated with hospitalization (Table 3).

To address whether serum possessed any suppressive activity on NK cells after surgery, PBMC from healthy volunteers were stimulated with *S. aureus* in the presence of the patients’ sera obtained before and after surgery. Sera of day 8 after surgery decreased the IFN-γ production by NK cells by 50% in comparison with pre-surgery sera. Sera from earlier time points did not affect the IFN-γ synthesis (Figure 7b). In contrast, none of the post-surgery sera had a significant impact on the expression of the IL-12Rβ2 chain on NK cells from healthy volunteers (Figure 7c). Thus, basal levels of circulating GDF-15 before surgery is associated with reduced IFN-γ synthesis by NK cells and with the length of hospitalization after surgery but is distinct to a circulating factor that inhibits NK cells late after surgery.

## 4. Discussion

In the present study, we show that invasive surgical tissue injury impairs the responsiveness of human circulating NK cells to *S. aureus*, a major cause of nosocomial infections. NK cells continuously lost their ability to produce IFN-γ during the first week after surgery that was independent of monocyte-derived IL-12 secretion. IFN-γ synthesis was minimal on day 8 and was associated with decreased expression of the IL-12 receptor and with the presence of a suppressive circulating factor. Treatment of the cells with recombinant IL-12 could largely restore the function of NK cells.

NK cells displayed a suppressed IFN-γ synthesis that was first apparent on day 4 after invasive spine surgery and persisted at least until day 8. This is in contrast to NK cell suppression induced by severe traumatic injury that rapidly establishes within 24 h after the insult [13]. Visceral surgery likewise causes NK cell suppression within 24 h but NK cell function recovers by day 7 [16]. Thus, NK cell suppression seems to be a common consequence of major mechanically induced tissue injury, but the degree of NK cell suppression seems to depend on the extent of tissue damage. Interestingly, severe injury does not affect the cytotoxic function of NK cells [13] which indicates that tissue damage interferes with selected signaling pathways.

NK cell suppression correlated with a decreased expression of the β2 chain of the IL-12R and was associated with reduced expression and activation/phosphorylation of the transcription factors T-bet and STAT4, respectively. Both, T-bet and pSTAT4 contribute to *IFNG* and *IL12RB2* gene transcription. The decreased T-bet expression in conjunction with reduced STAT4 phosphorylation might limit the expression of IL-12Rβ2 that further restricts STAT4 activation. Altogether, the establishment of such a negative feedback-loop between T-bet/STAT4 activity and the IL-12Rβ2 might drive the development of NK cell suppression within the first week after surgery. Breaking this negative feedback-loop by artificially increasing the level of IL-12 during stimulation with *S. aureus* increased the expression of T-bet, restored the expression of IL-12Rβ2, and improved the IFN-γ synthesis of NK cells after surgery.

NK cell-derived IFN-γ synthesis is not only required for immune defense against bacterial infections but also contributes to the regeneration process after tissue damage [17,18]. Consequently, NK cell suppression after surgery might increase the susceptibility to infections and might delay tissue regeneration. Notably, none of the patients developed an infectious complication. We assume that the suppression of NK cell-derived IFN-γ production alone is not sufficient to cause nosocomial infections but that additional immune defense mechanisms such as monocyte function must be impaired. Monocyte deactivation is a hallmark of immune dysregulation after severe trauma and is reflected by the impaired capacity of the cells to produce TNF-α and IL-12 [12,19]. We did not find evidence for impaired cytokine synthesis of monocytes after invasive spine surgery which is in line with a previous study from Hensler et al. who showed that visceral surgery does not affect monocyte-derived IL-12 synthesis [20]. This finding provides a novel aspect in the understanding of the immunomodulation after severe tissue injury, i.e., that trauma-induced NK cell suppression may develop independent from monocyte deactivation.

We observed that supplementation with IL-12 and IL-2 at least partially reversed NK cell suppression after surgery and thus might be therapeutically used to improve the immune defense against invading pathogens. Of note, early after surgery when NK cell suppression was not yet established the addition of IL-12/IL-2 caused an exaggerated IFN-γ release by NK cells that might increase the risk for inflammatory complications. Such side effects of IL-12 are known for its use in cancer therapy [21]. NK cell activation is only required in case of pathogen invasion and should be avoided in the absence of infection. Instead of prophylactically boosting the immune system by inflammatory mediators, a more promising approach might be to prevent the development of NK cell suppression.

Reduced monocyte HLA-DR expression is a prognostic marker for enhanced risk for opportunistic infections [22,23]. So far, clinical studies that aimed to restore monocyte function after surgical or accidental trauma have not shown a clear benefit in terms of mortality [24]. We suggest that the restoration of HLA-DR alone might not be sufficient to improve the immune defense after severe injury. This assumption is based on the fact that post-traumatic infections are frequently caused by rapidly dividing bacteria such as *S. aureus*, *E. coli*, and *Pseudomonas aeruginosa*. The immune defense against such pathogens requires the immediate activation of innate immune cells (e.g., neutrophils and macrophages) that eradicate the pathogens, a process that does not depend on HLA-DR. Therefore, increasing HLA-DR expression on monocytes after trauma likely is of minor relevance during the early phase of the immune response against opportunistic infections. Instead, sensing and elimination of pathogens is of major importance. The bactericidal capacity of phagocytes is improved by diverse cytokines (e.g., IFN-γ, TNF-α, GM-CSF) that are released by NK cells [25]. Therefore, we propose, that novel therapies that integrate the restoration of NK cell-derived cytokine secretion after severe injury might improve the function of phagocytes including monocytes/macrophages and neutrophils and consequently ameliorate the resistance against nosocomial infections.

We recently identified GDF-15 as a circulating factor after severe injury that is associated with NK cell suppression early after injury and with an enhanced risk for nosocomial infections [13]. In contrast to the fulminant and long-lasting release of GDF-15 after severe trauma, the concentration of GDF-15 after surgery that we observed here was only transiently increased. The signals that triggered the release of GDF-15 after surgery are still unknown and seemed to be independent from the extent of tissue injury (indicated by circulating myoglobin), CRP, the duration of surgery, or from blood transfusion. According to the low levels of GDF-15 after surgery, the patients’ sera did not possess an inhibitory capacity on NK cells of healthy donors early after surgery except for serum obtained on day 8 that led to a diminished production of IFN-γ. Nevertheless, GDF-15 levels before surgery correlated with low IFN-γ synthesis of NK cells both, before and after surgery and with the duration of hospitalization. Therefore, we assume that GDF-15 might be useful as a prognostic marker to identify patients with low NK cell activity and with an enhanced risk for post-surgical complications, but GDF-15 was not directly responsible for the decreased NK cell function after surgery. Similarly, GDF-15 has proven successful in the identification of patients at increased risk for acute kidney injury after cardiac surgery [26].

Due to the design as a pilot study, only few patients were included. As none of the patients developed an infectious complication, it is unfeasible to estimate the risk for such complications depending on NK cell function and GDF-15 levels. A larger follow-up study is required to address this issue.

## 5. Conclusions

Major musculoskeletal surgery suppresses the capacity of NK cells to produce IFN-γ upon exposure to *S. aureus* which develops independent from monocyte deactivation. NK cell suppression is associated with diminished transcription factor expression and correlates with the pre-operative level of circulating GDF-15. The quantification of GDF-15 on admission might identify patients at increased risk for post-surgical complications.

## Figures and Tables

**Figure 1 life-12-00013-f001:**
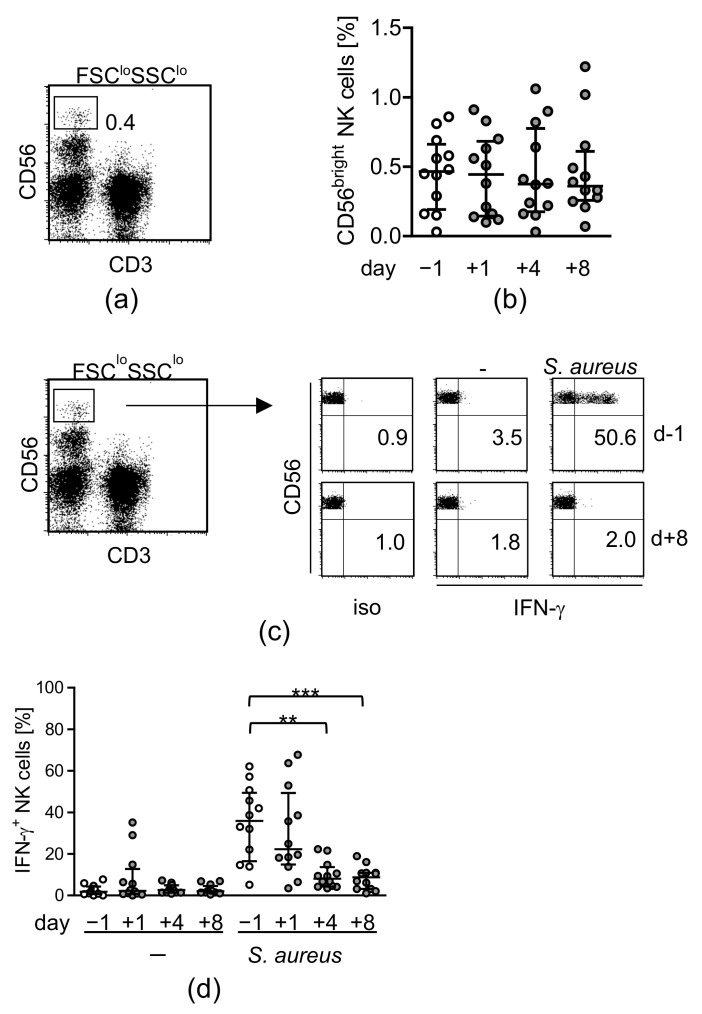
NK cells lose their capacity to respond to *S. aureus* during recovery from major surgery. PBMC were isolated before (d − 1) and at different time points after surgery (d + 1, d + 4, d + 8) and were analyzed by flow cytometry. (**a**) CD56^bright^ NK cells were gated as CD3-CD56^bright^ cells within the FSC^lo^SSC^lo^ lymphocyte gate. (**b**) Cumulative data on the frequency of CD56^bright^ Nk cells among lymphocytes. (**c**) PBMC were cultured in the absence (-) or presence of inactivated *S. aureus* and intracellular IFN-γ was detected by flow cytometry. Dot plots depict the gating strategy of CD56^bright^ NK cells from one representative patient before and 8 d after surgery. The threshold for positive staining was set according to the isotype antibody control staining (iso). Numbers in the lower right quadrant indicate the percentage of IFN-γ-producing cells among total CD56^bright^ NK cells. (**d**) Cumulative data on the frequency of IFN-γ^+^ NK cells. Statistically significant differences were tested using Wilcoxon signed rank test. **, *p* < 0.01; ***, *p* < 0.001.

**Figure 2 life-12-00013-f002:**
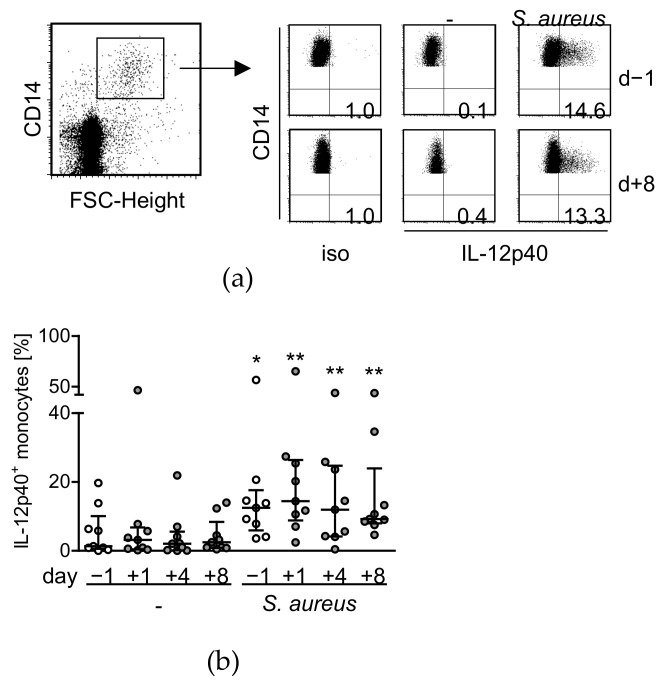
Monocytes are not impaired in IL-12p40 synthesis after major surgery. PBMC were isolated before (d − 1) and at different time points after surgery (d + 1, d + 4, d + 8) and were cultured in the absence (-) or presence of *S. aureus*. The synthesis of IL-12p40 was analyzed by intracellular flow cytometry. (**a**) Dot plots depict the gating strategy of CD14^+^FSC^int^ monocytes and the determination of IL-12p40 expression from one representative patient before and 8 d after surgery. The threshold for positive IL-12p40 staining was set according to the isotype antibody control staining (iso). Numbers in the lower right quadrant indicate the percentage of IL-12p40-producing cells among total monocytes. (**b**) Cumulative data on the frequency of IL-12p40^+^ monocytes. No statistically significant differences were detected between the activity of monocytes before and after injury. Asterisks denote significant differences between unstimulated (-) and stimulated (*S. aureus*) monocytes as calculated using the Wilcoxon signed rank test. *, *p* < 0.05; **, *p* < 0.01.

**Figure 3 life-12-00013-f003:**
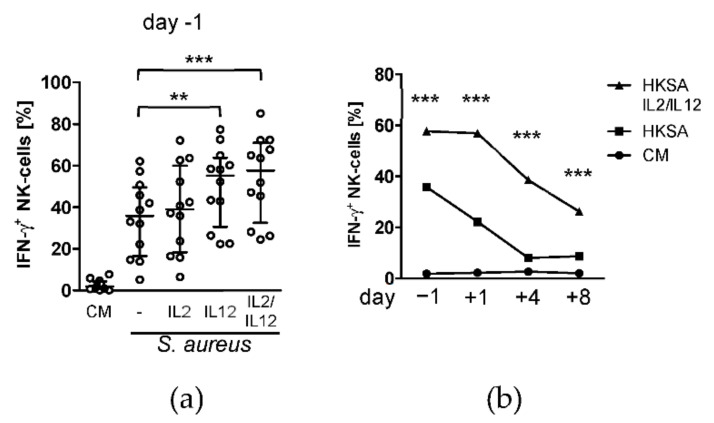
NK cell suppression after surgery can be restored by supplementation with IL-12. PBMC were isolated before (d − 1) and at different time points after surgery (d + 1, d + 4, d + 8) and were stimulated with inactivated *S. aureus* in the absence (-) or presence of recombinant IL-2, IL-12, or a combination of both. The synthesis of IFN-γ in CD56^bright^ NK cells was determined by intracellular flow cytometry. Cells cultured in the absence of bacteria and cytokines served as negative control (CM, culture medium). (**a**) The scatter plot shows the individual values of all patients the day before surgery (day −1). The values for unstimulated cells and cells cultured in the absence of cytokines are also part of Figure 1d. (**b**) Median values of the percentage of IFN-γ^+^ NK cells from all patients at each time point. For reason of clarity, only results of the combined treatment with IL-2/IL-12 but not of the single cytokines are shown. The complete data set of all culture conditions are shown in Appendix A. Statistically significant differences between absence and presence of supplemental cytokines were tested using Wilcoxon signed rank test. **, *p* < 0.01; ***, *p* < 0.001. CM, culture medium only; HKSA, heat-killed *S. aureus*.

**Figure 4 life-12-00013-f004:**
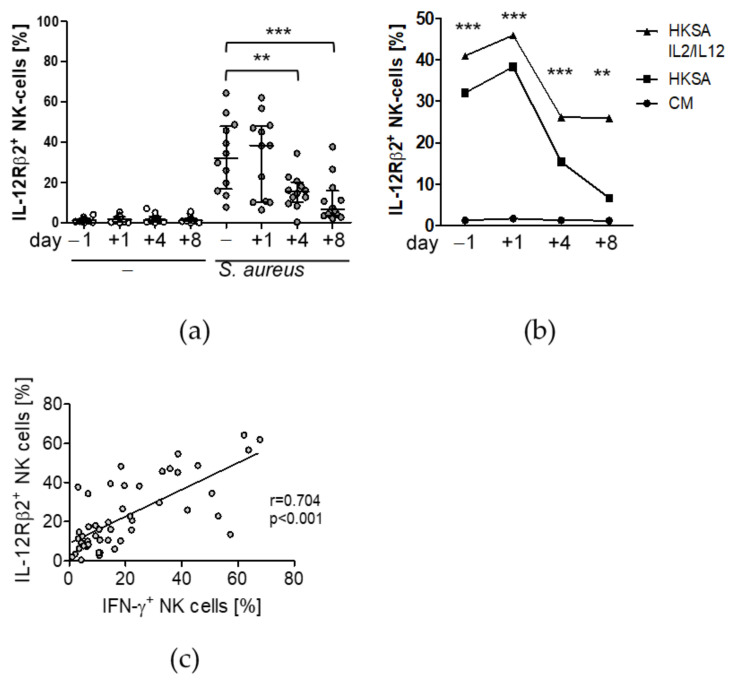
NK cell suppression after surgery correlates with the expression of the IL-12Rβ2 chain. PBMC were isolated before (d − 1) and at different time points after surgery (d + 1, d + 4, d + 8) and were cultured in the absence (-) or presence of inactivated *S. aureus*, the latter in combination with recombinant IL-2 and IL-12 as indicated. The expression of the IL-12Rβ2 chain on CD56^bright^ NK cells was determined by flow cytometry. Cells cultured in the absence of bacteria and cytokines served as negative control (CM, culture medium). (**a**) The scatter plot shows the individual values of IL-12Rβ2 expression for all patients at each time point. (**b**) Median values of the percentage of IL-12Rβ2^+^ NK cells from all patients at each time point. For reason of clarity, only results of the combined treatment with IL-2/IL-12 but not of the single cytokines are shown. The complete data set of all culture conditions are shown in Appendix A. Statistically significant differences between absence and presence of supplemental cytokines were tested using Wilcoxon signed rank test. (**c**) Spearman correlation of the percentages of IFN-γ^+^ and IL-12Rβ2^+^ NK cells after exposure to *S. aureus*. **, *p* < 0.01; ***, *p* < 0.001. CM, culture medium only; HKSA, heat-killed *S. aureus*.

**Figure 5 life-12-00013-f005:**
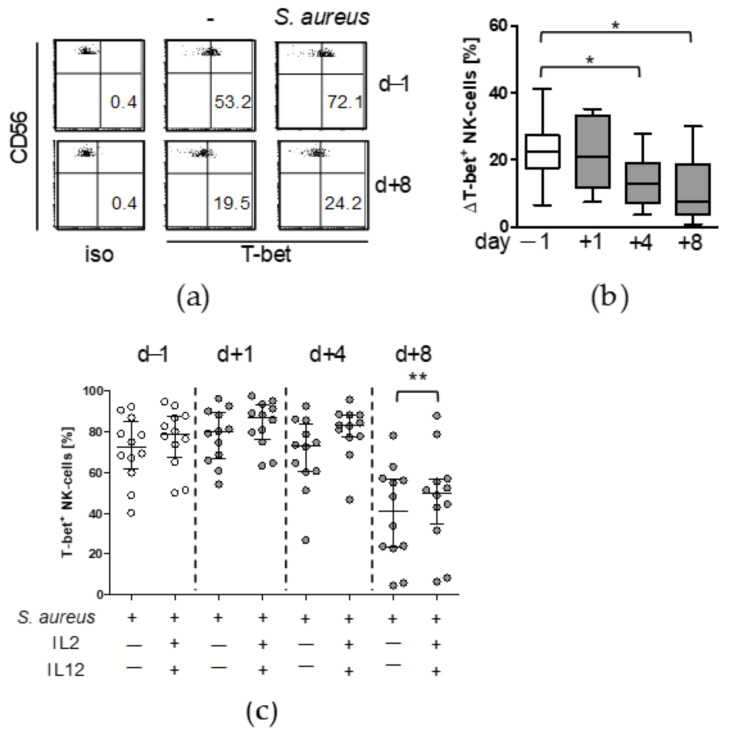
NK cell suppression after surgery is associated with reduced T-bet expression. PBMC were isolated before (d − 1) and at different time points after surgery (d + 1, d + 4, d + 8) and were cultured in the absence (-) or presence of inactivated *S. aureus*, the latter in combination with recombinant IL-2 and IL-12 as indicated. The expression of T-bet in CD56^bright^ NK cells was determined by intracellular flow cytometry. (**a**) Dot plots show the determination of T-bet expression from one representative patient before and 8 d after surgery. The threshold for positive T-bet staining was set according to the isotype antibody control staining (iso). Numbers in the lower right quadrant indicate the percentage of T-bet^+^ cells among total CD56^bright^ NK cells. (**b**) Cumulative data on the increase in T-bet expression after stimulation with *S. aureus* for all patients and time points. ΔT-bet was calculated as the difference between stimulation with *S. aureus* and unstimulated cells. Whiskers box plots show the median with interquartile range and min/max values. (**c**) Percentage of T-bet^+^ cells stimulated with *S. aureus* in the presence or absence of IL-2/IL-12. Scatter plot shows individual values. Horizontal lines indicate the median/interquartile range. Statistically significant differences were tested using Wilcoxon signed rank test. *, *p* < 0.05; **, *p* < 0.01.

**Figure 6 life-12-00013-f006:**
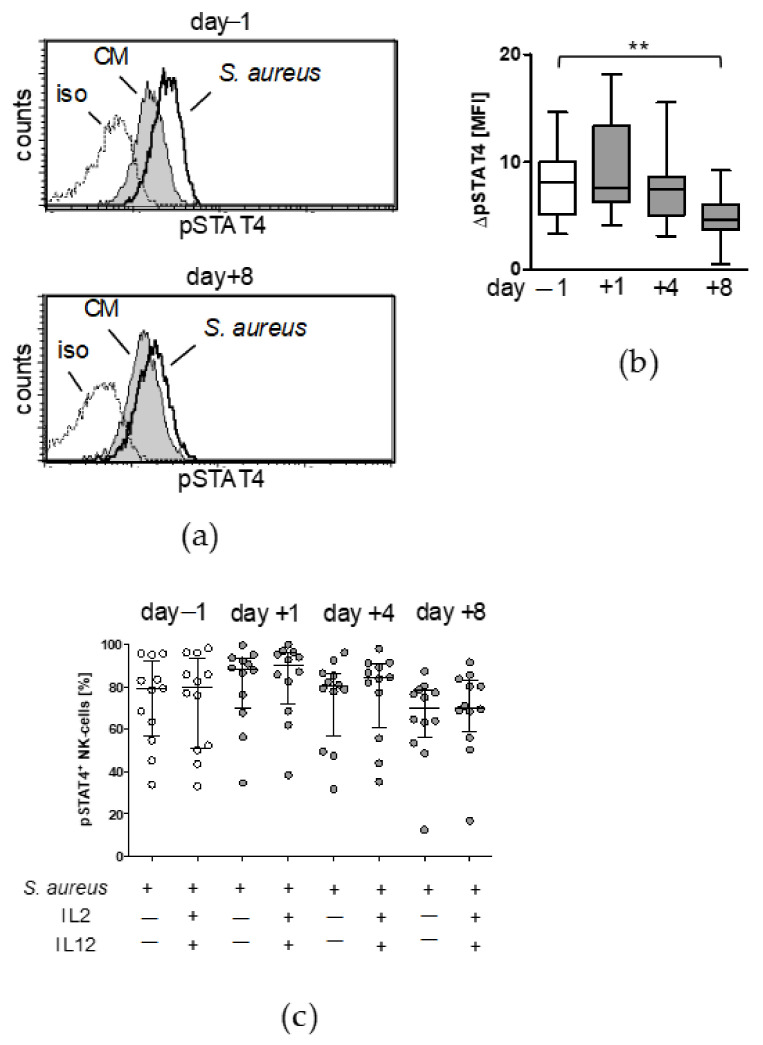
NK cell suppression is paralleled by reduced STAT4 activation late after surgery. PBMC were isolated before (d − 1) and at different time points after surgery (d + 1, d + 4, d + 8) and were cultured in the absence (-) or presence of inactivated *S. aureus*, the latter in combination with recombinant IL-2 and IL-12 as indicated. The expression of activated STAT4 (pSTAT4) in CD56^bright^ NK cells was determined by intracellular flow cytometry. (**a**) Histograms show the overlay of the fluorescence intensities for pSTAT4 in the absence (CM, culture medium) and presence of *S. aureus* from one representative patient before and 8 d after surgery. The dotted line indicates the isotype antibody control staining (iso). (**b**) Cumulative data of the increase in the mean fluorescence intensity (MFI) after stimulation with *S. aureus* for all patients and time points. ΔpSTAT4 was calculated as the difference between stimulation with *S. aureus* and unstimulated cells. Whiskers box plots show the median with interquartile range and min/max values. (**c**) Percentage of pSTAT4^+^ cells stimulated with *S. aureus* in the presence or absence of IL-2/IL-12. Scatter plot shows individual values. Horizontal lines indicate the median/interquartile range. Statistically significant differences were tested using Wilcoxon signed rank test. **, *p* < 0.01.

**Figure 7 life-12-00013-f007:**
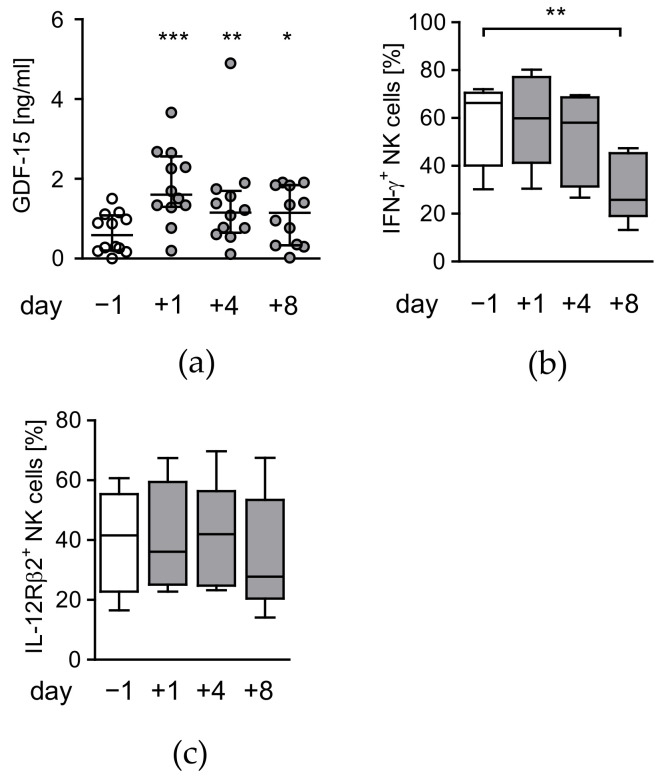
Increased levels of GDF-15 are not associated with NK cell suppression after surgery. (**a**) Individual levels of GDF-15 in the serum before (d − 1) and at different time points (d + 1, d + 4, d + 8) after surgery. Horizontal lines indicate the median/interquartile range). (**b**,**c**) PBMC from healthy donors (n = 5) were cultured in the serum obtained from patients at different time points (d − 1 to d + 8) and stimulated with *S. aureus*. The expression of IFN-γ (**a**) and the IL-12Rβ2 (**b**) were determined by flow cytometry. Statistically significant differences were tested using Wilcoxon signed rank test. *, *p* < 0.05; **, *p* < 0.01; ***, *p* < 0.001.

**Table 1 life-12-00013-t001:** Patient characteristics.

	Patients (n = 12)
age (y)	40 (19–67) ^1^
sex (m/f)	2/10
no. of fused segments	7 (3–11)
**co-morbidities**	
diabetes (n, %)	1/8.3
cardiovascular (n, %)	3/25
autoimmunity	2/16.6
smoker	0/0
duration of surgery (h)	6 (5–7)
blood transfusion (mL)	500 (0–1750)
ICU length of stay (d)	1 (1–2)

^1^ Continuous variables are expressed as median (interquartile range).

**Table 2 life-12-00013-t002:** Serum parameters and whole blood leukocytes. ^1^ Values are expressed as median (interquartile range). Statistical significance between two days or versus the normal value of healthy individuals was tested using the Wilcoxon signed rank test. ***, *p* < 0.001 versus d − 1; ###, *p* < 0.001 versus the reference value of CRP (<0.5 ng/dL), IL-6 (<15 pg/mL), myoglobin (<110 ng/mL).

	Patients (n = 12)
Leukocytes (x103/µL)	
d − 1	7.9 (6.3–10.4) ^1^
d + 1	10.4 (7.5–12.5) ***
CRP (ng/dL)	
d + 1	4.4 (4.1–5.7) ^###^
d + 2	18.4 (14.6–22.0) ^###^
d + 4	9.1 (5.3–17.7) ^###^
IL-6 (d + 1) (pg/dL)	212 (118–273) ^###^
myoglobin (d + 1) (ng/mL)	928 (557–1693) ^###^

**Table 3 life-12-00013-t003:** GDF-15 levels before surgery correlate with low NK cell activity and hospitalization.

Correlation of GDF-15 on d − 1 with	Spearman r	*p*-Value
**IFN-γ d − 1**	−0.7622	**0.006**
IFN-γ d + 1	−0.5455	0.071
**IFN-γ d + 4**	−0.6573	**0.024**
IFN-γ d + 8	−0.2727	0.391
**LOS** ^1^	0.6679	**0.021**

^1^ LOS, length of stay. Significant correlation is marked with bold.

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
