# Peer review of "Major Surgical Trauma Impairs the Function of Natural Killer Cells but Does Not Affect Monocyte Cytokine Synthesis"

_life, 2021, doi:10.3390/life12010013_

Round 1

Reviewer 1 Report

The manuscript submitted by the Roman Maximilian Müller-Heck et.al, is a well-written paper, however similar studies evaluating the role of IL12 induced production of IFNg the in context of other  infections or different cell type have been reported/discussed earlier, limiting the novelty of the study.

Comments-

  1. Did the authors evaluate the T cells phenotype/function as they are also involved in IFNg secretion. 
  2. Do the authors suspect similar effect on CD8+ T cytotoxic cells?
  3. Did the authors look at expression of other markers like perforin or Granzyme B or performed assays to determine reduced NK cell function including cytoxicity assays, GRA to determine the activity of NK cells?
  4. Can the authors comment on or cite references that discusses transcription factors of Interferon gamma in NK cells? The reference cited in the manuscript is with regard to T cells.
  5. Did the authors get a chance to look at levels of GDF-15 in patients with NK cell immunodeficiencies like  X-SCIDs or HLH. Is this biomarker associated with severe injury or with low NK cell function?

Reviewer 2 Report

The manuscript entitled „Major surgical trauma impairs the function of natural killer cells but does not affect monocyte cytokine synthesis” by Muller-Heck et al. is of great interest for the readers of “Life”.

The authors analysed the impact of invasive spine surgery on Staph. aureus induced IFN-synthesis by NK cells which drops significantly and was associated with decreased IL-12-receptor expression. However, NK-cell function could partly be restored by the addition of IL-12. Interestingly, preoperative level of GDF-15 was correlated with the extend of NK cell suppression.

Comments:

The subject of this manuscript is of high clinical relevance since immune suppression in the course of (surgical) trauma is generally seen as main reason for establishment of secondary infections during hospitalization which can be a life threatening risk for the patients. This observational studies proves evidence that NK cells were severely depressed and did not recover over a long period. Mechanistic analysis showed that those cells partially lost their ability to react to IL-12. This is probably due to reduced IL-12 receptor expression which might be due to decreased expression of the transcription factors STAT-4 and T-bet. An original study is provided as proved by pubmed analysis. The manuscript itself is clearly structured and the methods were described in a reproducible manner. The presentation of the results is adequate and the diagrams, tables and legends were without mistakes to a large extend. The discussion is well structured.

However, some minor questions and issues remain as pointed out in the following.

  1. Patients span a great age range, and so does the IFN-y-synthesis capacity by NK-cells (Fig.1). Is a correlation between age and degree of NK-cell depression observeable?
  2. Figure 1A: Do you have analysed only CD56bright cells? Does this mean the CD56mid cells were not analysed or do they not produce IFN-y?
  3. Figure 2, legend: A figure 2D is mentioned and described but not shown. It is stated that no statistical significant differences were detectable but there are some differences indicated in Figure 2B. Please correct the legend.
  4. IL-2 did not stimulate NK-cells: Did you check also for IL2-receptor expression on NK cells?
  5. Figure 3b: please define abbreviations (HKSA, CM) additionally in the legend to the figure (same for Figure 4).
  6. Have you performed experiments such as pre-cultivation of NK cells from healthy volunteers with GDF-15 followed by analysis of Staph. aureus stimulated IFN-y synthesis? If yes (and not published yet), it might support your hypothesis that GDF-15 alters the reaction of NK cells and would further improve your (already very good) manuscript.
